# Effectiveness of Hospital Fit on Physical Activity in Hospitalized Patients: A Stepped-Wedge Cluster-Randomized Trial and Process Evaluation

**DOI:** 10.3390/s24185920

**Published:** 2024-09-12

**Authors:** Hanneke C. van Dijk-Huisman, Niek Koenders, Rik G. J. Marcellis, Indy G. M. Smits, Thomas J. Hoogeboom, Ton A. F. Lenssen

**Affiliations:** 1Department of Physical Therapy, Maastricht University Medical Center, 6229 HX Maastricht, The Netherlands; hanneke.huisman@mumc.nl (H.C.v.D.-H.); rik.marcellis@mumc.nl (R.G.J.M.); 2Department of Rehabilitation, Radboud University Medical Center, 6525 GA Nijmegen, The Netherlands; niek.koenders@radboudumc.nl (N.K.); indy.smits@radboudumc.nl (I.G.M.S.); thomas.hoogeboom@radboudumc.nl (T.J.H.); 3IQ Health Science Department, Radboud University Medical Center, 6525 GA Nijmegen, The Netherlands; 4CAPHRI School for Public Health and Primary Care, Maastricht University, 6211 LK Maastricht, The Netherlands

**Keywords:** activity monitoring, walking, effectiveness, mHealth, physical activity, physiotherapy, wearable sensors

## Abstract

This study investigates the effectiveness of using Hospital Fit as part of usual care physiotherapy on the physical activity (PA) behavior of hospitalized patients compared to patients who received physiotherapy before implementation of Hospital Fit. In addition, a process evaluation is conducted. A prospective, multi-center, mixed-methods stepped wedge cluster randomized trial was performed at the cardiology and medical oncology departments of two Dutch university medical centers. Patients were included in the non-intervention or intervention phase. During the non-intervention phase, patients received usual care physiotherapy. During the intervention phase, Hospital Fit was additionally used. Mean time spent walking, standing, lying/sitting per day and the number of postural transitions from lying/sitting to standing/walking positions were measured using an accelerometer and analyzed using linear mixed models. A process evaluation was performed using questionnaires and semi-structured interviews with patients and focus-group interviews with healthcare professionals. A total of 77 patients were included, and data from 63 patients were used for data analysis. During the intervention phase, the average time spent walking per day was 20 min (95% confidence interval: −2 to 41 min) higher than during the non-intervention phase (*p* = 0.075). No significant differences were found for mean time spent standing per day, mean time spent lying/sitting per day, or the number of postural transitions per day either. During the intervention phase, 87% of patients used Hospital Fit at least once, with a median daily use of 2.5 to 4.0 times. Patients and healthcare professionals believed that Hospital Fit improved patients’ PA behavior and recovery. Insufficient digital skills and technical issues were described as challenges. Although patients and healthcare professionals described Hospital Fit as an added value, no statistically significant effects were found.

## 1. Introduction

During a hospital stay, patients spend little time standing and walking, and prolonged periods of sedentary behavior are common [1,2]. Low physical activity (PA) has been associated with negative health outcomes, such as functional decline [3,4], increased length of hospital stay (LOS) [5], increased risk of institutionalization [1,6] and mortality [7]. Low PA during a hospital stay does not only occur in mobility-dependent patients and walking aid users [8]. It is found in patients of all ages, surgical and nonsurgical populations, highlighting the severity and magnitude of the physical inactivity epidemic [9,10].

Previous research has shown that the risk of negative health outcomes can be reduced by improving patients’ PA levels. Interventions incorporating accelerometers have been shown to be effective in improving PA levels of hospitalized patients [11]. Hospital Fit (Hospital Fit 2.0, HFITAPP0, release 05, Maastricht Instruments B.V., Maastricht, The Netherlands) is a multimodal intervention designed to improve PA levels and functional recovery in hospitalized patients. It consists of an accelerometer and an accompanying smartphone application, which can be incorporated into the physiotherapy treatment [12]. Hospital Fit enables continuous PA monitoring with real-time feedback and allows patients and physiotherapists to set personal walking goals. Moreover, it enables physiotherapists to offer tailored exercise programs, monitor the extent of functional recovery, and provide information on the importance of remaining as active as possible. Additionally, it enables linking the data to the electronic medical record, making it available to other healthcare professionals. The accelerometer algorithm used in Hospital Fit has been validated to differentiate between sedentary, standing and dynamic activities, as well as to detect postural transitions in hospitalized patients [13].

The effectiveness of Hospital Fit as part of physiotherapy treatment has been investigated in two previous studies. A study investigated the potential of using Hospital Fit during the postoperative physiotherapy treatment of patients undergoing orthopedic surgery. Using Hospital Fit resulted in a mean increase of 28.4 min (95% confidence interval (CI): 5.6–51.3) standing and walking on postoperative day one compared with usual care physiotherapy [12]. This study was followed by a randomized controlled clinical trial investigating the effectiveness of using Hospital Fit during the physiotherapy treatment of patients hospitalized at internal medicine or pulmonology departments. While no statistically significant effects were observed, a trend was shown in favor of patients using Hospital Fit. The effect became more pronounced when Hospital Fit was used for a longer period, with a mean increase of 27.4 min (95% CI: −2.4–57.3) standing and walking on day five and 29.2 min (95% CI: −6.4–64.7) on day six of Hospital Fit use compared to patients receiving usual care physiotherapy [14]. Although both studies indicate that Hospital Fit may have potential to improve PA in hospitalized patients, it is unknown whether the results are generalizable to other patient populations and settings.

Therefore, the objective of this study is to investigate the effectiveness of using Hospital Fit as part of usual care physiotherapy on the PA behavior of patients admitted to the medical oncology or cardiology departments of the Maastricht University Medical Center (MUMC+) and Radboud University Medical Center (Radboudumc) compared to patients who received physiotherapy before the implementation of Hospital Fit. To allow a correct interpretation of the effectiveness of Hospital Fit, a process evaluation was performed additionally through investigating the reach, efficacy, adoption, implementation and maintenance of Hospital Fit in patients and healthcare professionals.

## 2. Materials and Methods

### 2.1. Design and Registration

A prospective, multi-center, mixed-methods stepped wedge cluster randomized trial (SW-CRT) and process evaluation was performed at the departments of cardiology and medical oncology of the MUMC+ Maastricht and Radboudumc Nijmegen, The Netherlands. Data were collected at four departments (clusters) between August 2022 and March 2023 (Table 1). All clusters began with a non-intervention phase during which patients received usual care physiotherapy. After two months, the first cluster transitioned to an implementation phase during which Hospital Fit was implemented in usual care physiotherapy. The Implementation of Change model by Grol and Wensing was used to support the implementation [15]. During the implementation phase, no patients were enrolled in the study. After one month, the respective cluster progressed to the intervention phase, during which Hospital Fit was embedded in usual care physiotherapy. Every subsequent month, another cluster followed. Inclusion in the study ended after the last cluster had spent two months in the intervention phase. In addition, a process evaluation was performed during the implementation and intervention phase using questionnaires, semi-structured interviews and focus-group interviews. Ethics approval was granted by the Medical Ethics Committee azM/UM (METC21-080). The study was performed in accordance with the Declaration of Helsinki. The study was registered at ClinicalTrials.gov: NCT05378724.

### 2.2. Participants

Patients undergoing physiotherapy while admitted to the department of medical oncology or cardiology were recruited during the week by their physiotherapist and asked for consent to be contacted by a researcher. The researcher informed potential participants about the study, waited 24 h to encourage voluntary participation, and then asked potential participants to sign informed consent to participate in the study. Confidentiality in data processing and patient anonymity were ensured. Eligibility criteria are outlined in Table 2.

### 2.3. Randomization and Blinding

Randomization of the cluster order was performed using four sealed, opaque envelopes, each containing a cluster name. Envelopes were opened by blinded staff members of the department of physiotherapy MUMC+, resulting in the following order: 1. Medical Oncology Radboudumc; 2. Cardiology MUMC+; 3. Cardiology Radboudumc; and 4. Medical Oncology MUMC+. Due to the nature of the intervention, patients, physiotherapists and researchers were not blinded to the study intervention.

### 2.4. Intervention and Study Procedures

#### 2.4.1. Regular Physiotherapy Treatment

Physiotherapy treatment was aimed at improving PA levels and enhancing functional independence in activities of daily living. The specific content of the physiotherapy sessions was tailored to each patient’s diagnosis and individual needs. During the non-intervention phase, patients received regular physiotherapy treatment and had their PA levels monitored with a MOX accelerometer. Patients and healthcare professionals did not use Hospital Fit and did not receive feedback on patients’ PA behavior.

#### 2.4.2. Hospital Fit

During the intervention phase, Hospital Fit was part of regular physiotherapy treatment. Hospital Fit consists of a smartphone app that is linked to a MOX accelerometer via Bluetooth connectivity and contains multiple functionalities (Figure 1) [13]. The app enables real-time PA monitoring through providing patients and physiotherapists insight into patients’ PA behavior (i.e., time spent walking, standing, lying/sitting and the number of postural transitions from lying/sitting to standing/walking positions per day). In addition, Hospital Fit contains the option of setting a daily walking goal (min.). This provides patients and healthcare professionals with a conversational tool for setting and evaluating PA goals. The algorithm of the MOX accelerometer is developed specially to measure the PA behavior of hospitalized patients. Validation showed sensitivity, specificity and accuracy values above 89.0% and a percentage error and an absolute percentage error below 8.0% [13]. PA data are transferred to the electronic medical record, making the data available to the multidisciplinary team. Furthermore, a recovery overview provides patients insight into their extent of functional recovery, using the modified Iowa Level of Assistance Scale (mILAS) [16]. Daily, the physiotherapist rated, using the mILAS-scores, the amount of assistance and type of walking aid needed to transfer from a supine position to sitting and vice versa, sit-to-stand, walking and climbing stairs. In the app, mILAS-scores are transformed to percentages (0% = dependent, 100% = independent) to visualize the level of independence. The mILAS has shown high validity, reliability and responsiveness when used to assess functional independence in hospitalized patients [16]. Additionally, the physiotherapist created and evaluated a personalized exercise program in the app, aimed at enhancing upper and lower limb strength, functional recovery and physical fitness. Moreover, the app contains a video explaining the importance of remaining active during hospitalization. Lastly, automatically generated notification messages are sent four times a day as a reminder to use Hospital Fit. A more detailed description of the functionalities of Hospital Fit has been published previously [14].

During the first treatment, the physical therapist helped install the application on the patient’s smartphone and initiated a connection between the app and the accelerometer. The functionalities were explained, supported by a paper-based user manual. Patients were advised to use Hospital Fit as often as they preferred, but at least once per day. Nurses and physicians could monitor the PA behavior, walking goal and mILAS score in the electronic patient record and were able to evaluate this with the patient during daily rounds.

### 2.5. Outcomes

#### 2.5.1. Demographics

The following demographic variables were extracted from the electronic medical record: age (years), sex (male/female), number of physiotherapy sessions received during study participation (n), LOS (days) and walking aid use before admission.

#### 2.5.2. Physical Activity Behavior

The primary outcome measure was mean time spent walking per day, calculated as the total number of minutes (min.) walking divided by the total number of days PA. Secondary outcome measures included mean time spent standing per day (min.), mean time lying/sitting per day (min.) and mean number of postural transitions from sedentary (lying/sitting) to active (standing/walking) positions per day. PA was monitored with the MOX accelerometer (MOX; Maastricht Instruments B.V., The Netherlands) and a tri-axial accelerometer sensor (ADXL362; Analog Devices, Norwood, MA, USA) housed in a compact with waterproof casing measuring 35 × 35 × 10 mm and weighing 11 g. During the first physiotherapy session, the accelerometer was fixed to the right anterior thigh (10 cm above the patella) using a hypoallergenic patch. Monitoring ended after nine days or on the day of discharge, whichever came first. A full day of measurement was defined as a 24-h period that began or ended at midnight. Days with ≥20 h of recorded wear time were considered valid measurement days and were included in the data analysis.

#### 2.5.3. Process Evaluation

Outcome parameters of the process evaluation were selected based on the RE-AIM framework Reach (percentage of Hospital Fit users reached, patient characteristics of users and non-users), Efficacy (perceived effectiveness of Hospital Fit use on patients’ PA behavior and perceived positive or negative outcomes of Hospital Fit use), Adoption (frequency of Hospital Fit use per day, use of different functionalities per day), Implementation (perceived barriers and facilitators to Hospital Fit use) and Maintenance (expected long-lasting effect of Hospital Fit on patients’ PA behavior, factors influencing maintenance of Hospital Fit use) [17]. Reach and adoption were assessed using a patient-reported questionnaire offered daily to all patients during the intervention phase (Appendix A). Reach was calculated by the ‘total number of users’ (i.e., patients enrolled during the intervention phase that used Hospital Fit at least once) divided by the ‘total number of included subjects’ (i.e., all patients enrolled in the study during the intervention phase). Patient characteristics were explored for users and non-users. Adoption was calculated as the number (n) of times per day patients used Hospital Fit and the different functionalities. Efficacy, implementation and maintenance were evaluated using individual, semi-structured interviews with patients and focus group interviews with healthcare professionals. The interview guides are provided in Appendix A.

### 2.6. Sample Size

The sample size calculation was based on the models of Chow et al. [18]. A previous pilot study showed an expected increase of mean time spent walking per day of 30% [12]. With a power of 0.80 and an effect size of 0.5, the required sample was 114 patients in total. The effect of clustering was modeled using the calculations by Killip et al. [19], with k = 4 clusters and an ICC of 0.01, resulting in a required sample of 160 patients in total. Accounting for a 10% drop-out rate, we aimed to enroll 180 patients in total in this study (45 patients per cluster).

### 2.7. Data Analysis

Stochastic regression imputation with fully conditional specification was used to impute missing values in cases where ≥5% of the data was missing. The difference in mean time spent walking per day was analyzed using linear mixed model analysis. Fixed factors were group (non-intervention versus intervention) and step (eight steps, one for each month of the trial), and the random factor was cluster (department). Secondary outcomes were analyzed using the same procedure. Data were analyzed according to the intention-to-treat principle. For all statistical analyses, a significance level of *p* < 0.05 was established. Data were analyzed using SPSS (version 28.0.0.0; IBM Corporation, Armonk, NY, USA).

Qualitative data were analyzed using a generic thematic approach. One researcher (NK) familiarized himself with the data by reading the transcripts and listening to the audio recordings. After this, NK generated the initial codes, which were later verified by an independent researcher (HvD). Subsequently, both researchers independently generated themes from the initial codes. Consensus was found on the themes by discussion in online meetings. The themes were, ultimately, reviewed, defined and renamed by all authors during subsequent drafts of the manuscript.

## 3. Results

### 3.1. Study Participation

In total, 525 patients were screened for study participation, and 306 patients (58%) fulfilled the inclusion criteria (Figure 2). Finally, of these 306 patients, 77 (25%) patients were included, of which 10 (13%) patients dropped out due to death (*n* = 3), a decline in health status (*n* = 1), cognitive decline (*n* = 1), skin irritation (*n* = 1), removal of the accelerometer by the patient (*n* = 1), experiencing study participation as a burden (*n* = 1), being unable to download the Hospital Fit app (*n* = 1) and unknown reasons (*n* = 1). Of these patients, the data of five patients were partially available for analysis. Moreover, data were missing completely for nine patients (12%) because they were discharged from the hospital before a valid measurement could be collected. The data of 63 patients (82%) were used in the analysis.

A total of 190 days of data were collected from participants in the control group and 79 days from participants in the intervention group. PA data were missing for 25 measurement days (9%) in 10 patients for the following reasons: low battery of the accelerometer (*n* = 2), accelerometer disconnected (*n* = 2), too early removal of the accelerometer due to expected hospital discharge (*n* = 3), accelerometer lost (*n* = 1), a decline in health status (*n* = 1) or experiencing study participation as a burden (*n* = 1). After imputation, the data of all 63 patients were complete for analysis.

### 3.2. Demographics

The mean age of the 63 patients was 68 years (95% CI: 65–71 years). Twenty-one patients were female (33%). Patients’ characteristics are provided in Table 3.

### 3.3. Primary Outcomes

PA data were collected for a mean of 4.3 days (95% CI: 3.6–4.9 days). The results of regression analysis showed that the mean time spent walking per day was 35 min (95% CI: 31–39) in patients enrolled during the non-intervention phase and 71 min (95% CI: 59–83) in patients enrolled during the intervention phase (Appendix A). Linear mixed model analysis showed that when corrected for cluster and step, patients enrolled during the intervention phase spent 20 min (95% CI: −2–41 min) more walking per day than patients enrolled during the non-intervention phase (*p* = 0.075). Details are provided in Table 4.

### 3.4. Secondary Outcomes

Linear mixed model analysis showed that when corrected for cluster and step, no significant differences were found between the non-intervention and intervention phases regarding mean time spent standing per day, mean time spent lying/sitting per day, and the number of postural transitions per day (Table 5).

### 3.5. Process Outcomes

In total, seven interviews were conducted with patients that used Hospital Fit. In addition, two focus group interviews (*n* = 4 per focus group) and one individual interview were performed with healthcare professionals. Moreover, 15 patients (100%) in the intervention group partially completed the standardized questionnaires. Data was missing on 8 days (10%) for six patients.

#### 3.5.1. Reach

Thirteen patients (87%) were defined as users and used Hospital Fit at least once during their hospital stay. Two patients (13%) were defined as non-users and did not use Hospital Fit during the intervention phase. The patients’ characteristics are presented in Table 6.

#### 3.5.2. Efficacy

Patients believed that using Hospital Fit increased their PA behavior. They explained that continuous monitoring and reminders sent in the app made them more engaged in being physically active, both on a conscious and subconscious level. They felt interested in gaining insight into their own PA levels and recovery processes, and understanding the data provided a sense of control. Moreover, it motivated them to be more active, as they were aware that their achievements and progression were also visible to their healthcare professionals. Furthermore, patients expressed a sense of pride and excitement in accomplishing the walking goals set by their physiotherapists. Healthcare professionals emphasized that gaining real-time insight into patients’ PA data enabled them to monitor the effect of physiotherapy, discuss walking goals and motivate patients. Both patients and healthcare professionals believed that Hospital Fit contributed to enhanced recovery. They both believed that being more physically active resulted in enhanced physical fitness and independence.

“*Well, I think it [Hospital Fit] did have a positive effect on my recovery, because I get goals that motivate me. […] Yes, I did recover faster and felt more in control*.”  [patient 57817]

However, patients also described that not all functionalities of Hospital Fit were perceived to be effective. They explained that mILAS scores did not always change over time and were not up to date. As a result, the recovery overview that provides patients insight into their extent of functional recovery did not have added value. They also elaborated that the video explaining the importance of remaining active during hospitalization was too simplistic.

Healthcare professionals stressed the importance of selecting the right patient subgroups to use Hospital Fit in. They expect that patients with a longer LOS may benefit more from using Hospital Fit as they have enough time to use the app and monitor their progression. Moreover, they expect that younger patients may also benefit due to better digital skills. Additionally, they described that patients with moderate motivation to stay active may benefit most from using Hospital Fit. Patients that are highly motivated to remain active already have sufficient intrinsic motivation and benefit little from additional extrinsic motivation through Hospital Fit. On the other hand, patients with little motivation will probably not modify their PA behavior through using Hospital Fit.

“*I think in the end it will come down to selecting the right patient for the right app, because I think we will have patients who already are sufficiently physically active and who are already enthusiastic… then it might add a very little bit and they could also do without it [Hospital Fit]*.”  [physiotherapist]

#### 3.5.3. Adoption

Hospital Fit was used for a median of 2.5 to 4 times per day. Daily use described per functionality is shown in Table 7.

#### 3.5.4. Implementation

Patients and healthcare professionals described Hospital Fit as user-friendly. Wearing the accelerometer did not bother patients in any way, and they often forgot that they were wearing one. They wanted long-term tracking for ongoing rehabilitation monitoring. Healthcare professionals said that the accelerometer was easy to fixate. Downloading the app on their smartphone and activating Bluetooth posed challenges to some patients, as it required digital skills and remembering passwords. This could often be resolved by receiving assistance from their physiotherapist. However, the physiotherapist also described that they needed sufficient digital skills to be able to provide that assistance.

Challenges that influenced Hospital Fit use were patients lacking time to use Hospital Fit due to a full daily schedule and technical issues, such as problems with synchronizing the data and connectivity issues.

“*So, after two days I thought oh yes, I have to look at that thing [Hospital Fit] too and yes, then it did not synchronize anymore*.”[patient 41548]

Other challenges mentioned by physiotherapists were having to use their own smartphone, and a high smartphone battery consumption. Sometimes, using Hospital Fit during the physiotherapy treatment took more time than intended. Reasons that were described were installation difficulties, slow synchronization of data due to limited internet connection, accelerometer batteries that needed to be recharged and the time needed to explain an exercise program. However, healthcare professionals explained that time could also be gained, because Hospital Fit stimulated patients to be engaged in their recovery and more independently active. In general, physiotherapists perceived Hospital Fit to be of high value as it provided them additional treatment options, enabling them to objectively evaluate PA behavior and adopt their treatment. Moreover, patients also described Hospital Fit as having added value and mentioned that they would have liked to use Hospital Fit for a longer period after discharge.

The limited use of Hospital Fit by nurses and physicians was also discussed. A nurse said that she was not informed about Hospital Fit. Although nurses received training in Hospital Fit use prior to study initiation, some nurses described that they were unable to attend these training sessions due to evening shifts or nightshifts. Moreover, nurses and physicians were informed three weeks before the start of the intervention, so they described that knowledge and implementation momentum were lost in the meantime.

“*That of course is difficult if you want to inform the nurses at the Cardiology ward, that they all have rotating shifts, and you can never inform everyone of course.*”  [nurse]

#### 3.5.5. Maintenance

Healthcare professionals were optimistic about future Hospital Fit use, envisioning its integration into routine nursing interventions. To improve maintenance, both groups supplied suggestions for improvement, including features to improve social interaction between patients and healthcare professionals, enhancing the visualization of data through graphs, adding global positioning system tracking and adding step count or walking distance as additional PA outcome measures. Healthcare providers also recommended integrating cut-off values about PA into the electronic health record, so that abnormal PA behavior can be detected and acted upon.

## 4. Discussion

This study investigated the effectiveness of Hospital Fit in addition to usual care physiotherapy on improving the PA behavior of patients admitted to the medical oncology or cardiology departments of two Dutch University Medical Centers. Although patients enrolled during the intervention phase had a 20 min (95% CI: −2 to 41 min) higher mean time spent walking per day compared to patients enrolled during the non-intervention phase, the between-group effect was not statistically significant (*p* = 0.075). Moreover, mean time spent standing and lying/sitting per day, as well as the number of postural transitions per day, did not significantly differ between groups either. The process evaluation showed that during the intervention phase, 87% of patients used Hospital Fit at least once, with a median daily use of 2.5 to 4.0 times. The functionality that was used most often was the PA monitoring option, followed by the recovery overview. Both patients and healthcare professionals were convinced that Hospital Fit could improve patients’ PA behavior and recovery process. Healthcare professionals indicated that not all patient subgroups may benefit and that it is important to select the right patients. Factors that positively influenced Hospital Fit use were its perceived efficacy, the app’s user friendliness and that patients were not bothered by wearing the accelerometer. Challenges that may negatively influence the use of Hospital Fit are the required digital skills of patients and healthcare professionals, the occurrence of technical issues and the additional time investment required from physiotherapists. Despite challenges, both patients and healthcare professionals described that they were willing to use Hospital Fit within usual care, although suggestions for improvement were provided.

Two previous studies have investigated the potential of Hospital Fit to improve the PA behavior of hospitalized patients [12,14]. A pilot study showed that using Hospital Fit during the physiotherapy treatment of patients following total knee and hip replacement surgery resulted in a mean increase of 28.4 min (95% CI: 5.6–51.3) standing and walking on postoperative day one compared to usual care physiotherapy [12]. A subsequent randomized controlled trial showed that using Hospital Fit did not significantly increase the PA behavior of patients admitted to the department of pulmonology or internal medicine. However, despite the lack of significance, a trend was seen in favor of Hospital Fit use with a mean increase of 27.4 min (95% CI: −2.4–57.3) standing and walking on day five, and 29.2 min (95% CI: −6.4–64.7) on day six of Hospital Fit use compared to patients receiving usual care [14].

A recent systematic review and meta-analysis showed a small positive effect in favor of interventions using activity trackers during or after inpatient care (standardized mean difference: 0.34, 95% CI: 0.12–0.56) [20]. However, the authors concluded that the effectiveness of PA improves when the intervention is provided both during and after the inpatient period. This seems to be consistent with the finding from the current study, where a non-significant effect was found during the inpatient period, and where patients and healthcare professionals both indicated that it would have been of added value to continue using Hospital Fit after discharge.

The results of the current study should be interpreted with caution, because the intended sample size has not been achieved and there are significant differences in inclusion numbers between clusters. We chose to implement Hospital Fit in the physiotherapy treatment of patients admitted to the cardiology or medical oncology departments because we expected these patients to be physically inactive for most of the day, to have sufficient digital skills and to have a LOS that would permit them to use Hospital Fit for at least 3 to 4 days. Although these populations seemed ideal, the intended inclusion numbers were not achieved. We intended to include 180 patients (45 per cluster). Unfortunately, we included as few as 77 patients (43%), and only the data of 63 patients (35%) could be included in the analysis. Of these patients, only 15 patients (24%) were enrolled during the intervention phase in 4 clusters. Moreover, most patients (59%) were enrolled within the Cardiology MUMC+ cluster, which was the only cluster that met the expected inclusion numbers. In the other three clusters, inclusion numbers were not met. In the Cardiology Radboudumc cluster, only 53% of the expected patients were included, with only two patients enrolled during the intervention phase. In both medical oncology clusters, only nine patients could be included in 7 months (*n* = eight during the non-intervention phase, *n* = one during the intervention phase). Ultimately, only the data of seven patients were included in the analysis (*n* = 3 Medical Oncology Radboudumc cluster, *n* = 4 Medical Oncology MUMC+ cluster), with no data available from patients included during the intervention phase. The main reasons for the low inclusion numbers in the two medical oncology clusters were as follows: (1) the number of referrals to physiotherapy in these clusters was only 42% of that anticipated; (2) 26 (50%) of the 52 eligible patients were discharged within 1 to 2 days after referral to physiotherapy; (3) a smaller proportion of patients declined participation because they were not interested (*n* = 9), already had a lot on their mind (*n* = 1) or because family discouraged participation (*n* = 1). This raises the question of whether interventions aimed at enhancing the PA behavior of patients hospitalized at a medical oncology ward are needed. Some patients may not have required interventions to stay active during hospitalization, but it could also be possible that some patients were not referred to physiotherapy because their medical condition prevented them from increasing their PA behavior. Due to the low numbers of included patients, the skewed ratio between patients in the control and intervention groups and the uneven distribution between clusters, the results of this study do not provide an accurate representation of the effectiveness of Hospital Fit and thus may not be generalizable to other patients receiving physiotherapy during a hospital admission. Moreover, the low and skewed inclusion rates in the other three clusters may have biased the effectiveness of Hospital Fit on the PA behavior of patients in the Cardiology MUMC+ cluster. Unfortunately, the inclusion period could not be extended due to the chosen study design. The stepped-wedge cluster randomized trial (SW-CRT) was chosen as it provides a robust framework for evaluating the effectiveness of interventions in a real-world setting. It allowed for phased implementation of Hospital Fit across the different clusters, enabling sufficient time and staffing to support the implementation in each cluster. Moreover, all clusters eventually received the intervention, reducing ethical concerns about inequity and enabling an investigation of the effectiveness of Hospital Fit at a department level [21,22]. Classical and cluster randomized controlled trials were deemed less suitable in this study due to an increased risk of contamination between patients and pragmatic challenges. Although a SW-CRT has many advantages, in the current study, we also encountered the risks associated with the use of this design. Due to the SW-CRT design, the total study duration was predetermined and consisted of 7 months (steps), during which patients could be included. Every month, a cluster transitioned to the next step, regardless of whether inclusion numbers were met. As a result, the design did not allow us to extend the inclusion period when expected numbers were not met. Possible solutions would have been to let a cluster transition to the next step dependent on the number of inclusions (sample size) achieved rather than on time. However, this was not a solution for the current study, as the inclusion in the medical oncology clusters was progressing so slowly that it was not feasible to proceed. Moreover, this solution could result in multiple clusters entering the implementation phase simultaneously, leaving less attention per cluster devoted to the implementation.

## 5. Conclusions

This study investigated the effectiveness of Hospital Fit as part of regular physiotherapy treatment on PA behavior among patients admitted to cardiology or medical oncology departments of two Dutch University Medical Centers. Our analysis showed no significant differences in PA behavior because of Hospital Fit use. In total, 87% of patients used Hospital Fit, with a median usage of 2.5 to 4 times per day. While patients and healthcare professionals perceived benefits, they also described challenges such as digital skills and technical issues.

## Figures and Tables

**Figure 1 sensors-24-05920-f001:**
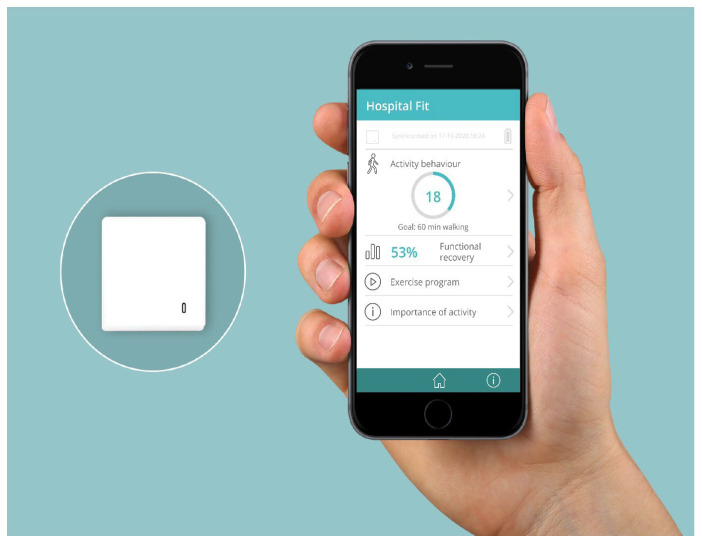
Hospital Fit app showing different functionalities (**right**) and MOX accelerometer (**left**).

**Figure 2 sensors-24-05920-f002:**
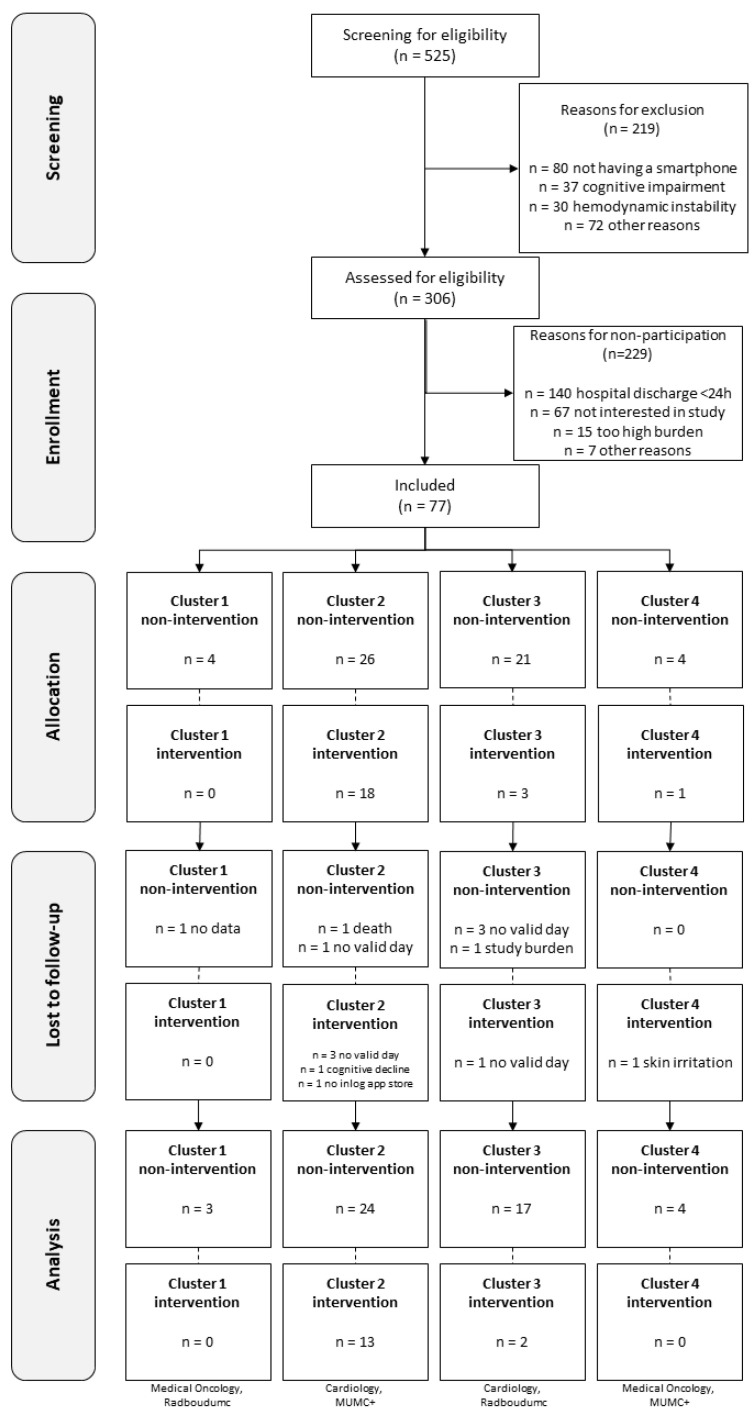
Consort flowchart with numbers of patients.

**Table 1 sensors-24-05920-t001:** Overview of the stepped-wedge cluster-randomized trial with four clusters (departments). MUMC+, Maastricht University Medical Center; Radboudumc, Radboud University Medical Center.

	August 2022	September 2022	October 2022	November 2022	December 2022	January 2023	February 2023	March 2023
**Cluster 1** Medical Oncology, Radboudumc	Non-intervention		Implementation	Intervention				
**Cluster 2** Cardiology, Maastricht UMC+	Non-intervention			Implementation	Intervention			
**Cluster 3** Cardiology, Radboudumc	Non-intervention				Implementation	Intervention		
**Cluster 4** Medical Oncology, Maastricht UMC+	Non-intervention					Implementation	Intervention	

**Table 2 sensors-24-05920-t002:** Inclusion and exclusion criteria for study participation.

**Inclusion Criteria**
Age ≥ 18 years
Receiving physiotherapy while hospitalized at the department of medical oncology or cardiology
Sufficient understanding of the Dutch language
Having access to a smartphone (operating system ≥ iOS13.0 or Android 8.0)
Able to use a smartphone application
Able to walk independently two weeks before admission, as scored on the Functional Ambulation Categories (FAC > 3)
**Exclusion Criteria**
Presence of a contraindication to walking (as reported by a medical specialist in the electronic medical record)
Presence of a contraindication to wearing an accelerometer on the upper leg, fixated by a hypoallergenic plaster
Admitted for cancers of the head and neck
Admitted with cardiac arrhythmia or hemodynamic instability requiring medication over 48 h or invasive treatment (e.g., pacemaker)
Mentally incapacitated subjects as reported by healthcare professionals in the medical record
Cognitive impairment (delirium/dementia) as reported in the medical record by a healthcare professional
Unable to participate in the informed consent procedure or unable to provide written informed consent
A life expectancy < 3 months
Previous participation in this study

**Table 3 sensors-24-05920-t003:** Characteristics of patients included in the study.

Characteristic	Patients	Non-Intervention Group	Intervention Group
Number of patients	63 (100%)	48 (76%)	15 (24%)
Age in years	68 (65–71)	69 (65–73)	64 (60–69)
Sex, female	21 (33%)	17 (35%)	4 (27%)
Physiotherapy treatments during admission	5.7 (4.8–6.6)	5.6 (4.5–6.7)	6.1 (4.5–7.8)
Walking aid users before admission	14 (22%)	10 (21%)	4 (27%)
Length of hospital stay in days	16 (14–19)	16 (13–19)	19 (14–23)
Discharge location			
-Home	46 (73%)	34 (71%)	12 (80%)
-Other hospital	8 (13%)	5 (10%)	3 (20%)
-Rehabilitation center	7 (11%)	7 (15%)	0 (0%)
-Nursing home	2 (3%)	2 (4%)	0 (0%)

Data are presented as mean (95% confidence interval)) or absolute numbers (percentage).

**Table 4 sensors-24-05920-t004:** Linear mixed model analysis analyzing the difference in mean time spent walking per day between patients in the non-intervention phase and patients in the intervention phase (Hospital Fit use).

**Linear Mixed Model** Outcome: Time walking in minutes per day Fixed factors: Group (non-intervention versus intervention) and Step (month 1–8) Random: Intercept Subject: Cluster					
	**B**	**Standard Error**	***p*-Value**	**95% Confidence Interval for B**	
				**Lower Bound**	**Upper Bound**
Intercept	26	6	0.004	11	40
Group	20	11	0.075	−2	41
Step	4	2	0.083	0	8

**Table 5 sensors-24-05920-t005:** Linear mixed model analysis secondary outcomes.

**Time standing in minutes per day** Fixed factors: Group (non-intervention versus intervention) and Step (month 1–8) Random: Intercept Subject: Cluster					
	**B**	**Standard Error**	***p*-Value**	**95% Confidence Interval for B**	
				**Lower Bound**	**Upper Bound**
Intercept	24	3	<0.001	18	31
Group	4	7	0.537	−9	17
Step	2	1	0.089	0	5
**Time lying/sitting in minutes per day** Fixed factors: Group (non-intervention versus intervention) Random: Intercept Subject: Cluster					
	**B**	**Standard Error**	***p*-Value**	**95% Confidence Interval for B**	
				**Lower Bound**	**Upper Bound**
Intercept	1204	24	<0.001	1123	1285
Group	−23	38	0.542	−97	51
**Number of postural transitions per day** Fixed factors: Group (non-intervention versus intervention) and Step (month 1–8) Random: Intercept Subject: Cluster					
	**B**	**Standard Error**	***p*-Value**	**95% Confidence Interval for B**	
				**Lower Bound**	**Upper Bound**
Intercept	31	3	<0.001	26	37
Group	10	5	0.064	−1	21
Step	0	1	0.990	−2	2

**Table 6 sensors-24-05920-t006:** Patients’ characteristics of users and non-users in the intervention phase.

User (U) or non-user (NU)	U	U	U	NU	U	U	U	U	U	NU	U	U	U	U	U
Cluster (1, 2, 3 or 4)	2	2	2	2	3	2	2	2	3	2	2	2	2	2	2
Age in years	62	61	57	72	78	78	57	61	56	52	59	74	65	62	72
Sex, female (F) or male (M)	M	M	M	M	F	F	M	M	M	F	M	F	M	M	M
Number of physiotherapy treatments	3	5	3	5	4	5	7	5	3	3	7	5	3	3	3
Walking aid before hospital admission, yes (Y) or no (N)	N	N	N	Y	Y	Y	N	N	N	N	N	N	Y	N	N
Length of hospital stay in days	8	31	9	26	28	17	23	15	15	14	18	30	9	25	10

**Table 7 sensors-24-05920-t007:** Median (interquartile range) use of different functionalities of Hospital Fit per day.

Functionality	Day 1	Day 2	Day 3	Day 4	Day 5	Day 6	Day 7
PA monitoring	4 (1–8)	4 (1–8)	4 (1–6)	2.5 (1–8)	3 (1–8)	4 (1–5)	2.5 (1–4)
Recovery overview	2 (1–3)	2 (1–6)	3 (1–3)	3 (1–4)	2 (1–2)	1.5 (1–2)	2 (1–3)
Exercise program	1 (1–4)	4 (0–4)	2 (1–4)	1 (0–1)	1 (0–1)	1 (1–2)	1 (0–1)
Importance of PA	1 (1–1)	1 (0–1)	1 (0–1)	1 (0–1)	1 (0–1)	1 (0–1)	0

Abbreviation: PA, physical activity.

## Data Availability

The raw data supporting the conclusions of this article will be made available by the authors on request.

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
