# Peer review of "Effectiveness of Hospital Fit on Physical Activity in Hospitalized Patients: A Stepped-Wedge Cluster-Randomized Trial and Process Evaluation"

_sensors, 2024, doi:10.3390/s24185920_

Round 1
Reviewer 1 Report
Comments and Suggestions for Authors
This manuscript reports on monitoring of physical activity among hospitalized patients using the "Hospital Fit" app in mobile phone. The app seems interesting and useful in the activity monitoring. In general, this paper can be suitable for publication in sensors. However, there are some questions to be addressed as below.
1. The paper mentioned the use of accelerometers to monitor patients' physical activity. Firstly, regarding accuracy, has the data collection from the accelerometers employed advanced algorithms to enhance the precision of activity recognition? Is it more accurate than existing technologies in distinguishing between different types of activities, such as walking, standing, and sitting? Additionally, have special measures been taken to ensure the reliability and validity of the data during the synchronization and processing phases?
2. While real-time monitoring and goal-setting are highlighted features of this application, one would like to know if these functionalities are novel enough compared to other solutions in the existing literature to make a significant difference in clinical practice.
3. It should be reasonable to explain how to select the volunteers from cardiology and oncology as the subjects in your study. Could you explain whether this choice was influenced by the unique aspects of physical activity patterns?
4. It is also suggested that the authors discuss the recovery process involved other physiological and psychological factors. In addition, would these factors potentially have a significant impact on the promotion of physical activity recovery and the monitoring of real-time data?
5. What are the crucial factors that influenced the use of the "Hospital Fit" app? The authors are suggested to show some details about these factors during the implementation.
6. It is mentioned that the sample size did not meet expectations, which is indeed a significant limitation that may affect the generalizability of the findings. Is it possible to show provide insights for clinical when the sample size is relatively small?
Reviewer 2 Report
Comments and Suggestions for Authors
The authors reported effectiveness of hospital fit on physical activity in hospitalized patients with a stepped-wedge cluster-randomized trial and process evaluation. Some issues are needed to be fixed here:
1. Data from 63 patients were analyzed here. Are they enough or adequate for Hospital Fit analysis?
2. Sometimes the patients can not express their feeling directly or accurately. How do you make sure that they are satisfied with these outcomes? Maybe you can not acquire the accurate evaluation depending on your device.
3. All you did is actually based on some collection data and further analysis. Did you compare your system with other reported literatures?
Comments on the Quality of English LanguageMinor editing of English language required.
Round 2
Reviewer 2 Report
Comments and Suggestions for Authors
This paper could be published as present form.